# A Review of Australian Tick Vaccine Research

**DOI:** 10.3390/vaccines9091030

**Published:** 2021-09-16

**Authors:** Ala E. Tabor

**Affiliations:** 1Queensland Alliance for Agriculture & Food Innovation, Centre for Animal Science, The University of Queensland, 306 Carmody Road, St. Lucia, QLD 4072, Australia; a.tabor@uq.edu.au; 2School of Chemistry and Molecular Biosciences, The University of Queensland, 68 Cooper Road, St. Lucia, QLD 4072, Australia

**Keywords:** Australia, tick vaccines, *Rhipicephalus*, *Ixodes*

## Abstract

Tick vaccine research in Australia has demonstrated leadership worldwide through the development of the first anti-tick vaccine in the 1990s. Australia’s Commonwealth Scientific and Industrial Research Organisation’s (CSIRO) research led to the development of vaccines and/or precursors of vaccines (such as crude extracts) for both the cattle tick and the paralysis tick. CSIRO commercialised the Bm86 vaccine in the early 1990s for *Rhipicephalus australis*; however, issues with dosing and lack of global conservation led to the market closure of Tick-GARD in Australia. New research programs arose both locally and globally. The Australian paralysis tick *Ixodes holocyclus* has perplexed research veterinarians since the 1920s; however, not until the 2000s did biotechnology exist to elucidate the neurotoxin—holocyclotoxin family of toxins leading to a proof of concept vaccine cocktail. This review revisits these discoveries and describes tributes to deceased tick vaccine protagonists in Australia, including Sir Clunies Ross, Dr Bernard Stone and Dr David Kemp.

## 1. Introduction

Ticks are hematophagous (bloodsucking) arthropods, which have evolved with animal hosts. There are 70 known tick species in Australia, and of these, 14 are soft ticks (Argasidae), and 56 are hard ticks (Ixodidae). Sixteen species are known to feed on humans and domestic animals, while the remaining species feed on wild mammals, birds and reptiles (reviewed by Barker et al., 2014) [1]. The hard tick species, which have the most economic impact on livestock and companion animals in Australia, are *Rhipicephalus australis* (the Australian cattle tick, formerly *Rhipicephalus (Boophilus) microplus*) and *Ixodes holocyclus* (paralysis tick).

The Australian cattle tick belongs to the cryptic species of *R. microplus*, which currently contains three species *R. australis*, *R. annulatus* and *R. microplus*, with three clades for the latter A, B and C [2,3]. *Rhipicephalus australis* has been found to be most similar to *R. microplus* Clade A, which appears to have a global distribution, including Asia, USA, Kenya and South America [2,4,5], and was introduced into northern Australia through the importation of infested Brahman cattle from Indonesia (Batavia) in 1872 [6]. Overall, the impact of the cattle tick on the global cattle industry was calculated at USD 22–30 billion per annum in 2016 and in Australia, approximately USD 160 million per annum in 2005 [7,8]. Although the Australasian species has been renamed as *R. australis*, this appears to be a geographic potentially continental species drift of the original species from India, which co-evolved with *Bos indicus* (tick resistant) breeds of cattle [9]. The global distribution and emigration of cattle worldwide perhaps have led to geographic separation of clades and species of *R. microplus* due to limited introductions, as reported for Australia described above [6,9]. Nonetheless, the species complex is distributed worldwide with far-reaching impacts on cattle production, as described.

Prior to the reinstatement of *R. australis* (Fuller 1899, [10]) in 2012–2014 [2,11], the species was previously referred to globally as *Boophilus microplus* and as *R. (B.) microplus* (reviewed by Barker et al. [1]). The University of Queensland (UQ) has led several relevant tick phylogenetic studies through Professor Stephen Barker’s research [1,12]. The microbiology research community requires a review of genus/species name changes or names of a newly discovered species by the International Committee on Systematics of Prokaryotes prior to publishing the accepted name on approved lists in the International Journal of Systematic and Evolutionary Biology. The journal also has strict guidelines for naming conventions and the use of sequence data (International Journal of Systematic and Evolutionary Microbiology—About | Microbiology Society (microbiologyresearch.org)). In parasitology, however, an author can suggest a new species or genus name; if the publication is accepted, the publication automatically sets the precedence for the use of a new genus/species name. Thus, the name changes for *B. microplus* have not been scrutinised by an international committee and, as such, remain controversial. For the purposes of this review, the accepted name of *R. australis* is used for the Australian research into the cattle tick, although the original publications refer to *B. microplus* or *R. (B.) microplus*.

The Australian paralysis tick *Ixodes holocyclus* is an indigenous tick species and is distributed along the east coast of Australia [13]. Unlike *R. australis*, *I. holocyclus* has remained the species name since it was first described by Neumann in 1899 [14]. Of the 892 species of ticks worldwide, approximately 27 species are evidenced as causing host paralysis, including two species in Australia, *I. holocyclus* and *Ixodes cornuatus* [15]. *Ixodes holocyclus* has been considered the most toxigenic tick species, producing a family of neurotoxins known as holocyclotoxins, which is currently the only toxin family to date described for a toxin-producing tick species [16]. Unlike the cattle tick *R. microplus*, which is a one host tick species specific to cattle to complete their life cycle, *I. holocyclus* has a three-host life cycle, including hosts, such as Australian wildlife, and humans, livestock and companion animals as primary and incidental hosts, respectively [13]. It has been reported that approximately 10,000–100,000 animals are affected annually, with 5% deaths in affected livestock and companion animal pets [17].

Approaches for the development of anti-tick vaccines have been quite variable, particularly for *R. microplus* and *R. australis* globally. One approach has been to target gut-associated ‘concealed antigens’, which are usually not recognised by the host during natural infestation; however, when used as a vaccine antigen, the host’s antibodies attack the tick gut, disrupting the life cycle [18]. One disadvantage is that these vaccines require several boosts each year as tick challenge does not boost host responses to these types of antigens. Other vaccine approaches have focused on secreted salivary antigens, which conversely are assumed to be recognised by the host during the natural challenge and, thus, may potentially boost vaccine responses [19]. This is particularly relevant for tick species whose tick stages feed on their hosts for shorter periods, i.e., *I. holocyclus* [17,20]; however, these approaches have also been applied in the discovery of *R. microplus* antigens [19].

Australia has led the world in the development of anti-tick vaccines. The first anti-cattle tick recombinant vaccine was developed by researchers from Australia’s National Science agency—the Commonwealth Scientific and Industrial Research Organisation (CSIRO) in the late 1980s [18]. Attempts to develop a canine anti-paralysis tick vaccine were also reported in the 1980s in Australia through research, also originally led by CSIRO [20]. This review reports Australian research capacity in the development of anti-tick vaccines throughout the 19th–20th Century with considerable efforts from the 1980s to date.

## 2. Review Approach

A search using the PubMed database was undertaken with the following terms: ((tick) AND (ticks)) AND (vaccine)) AND (Australia). This search returned 112 articles. Articles that were not related to the topic were deleted (Australian authors, un-related vaccines), resulting in 65 references. Several of these references were reviews, or out of scope for this review, or were published by the same group from which selected articles were highlighted. Following this scrutiny, 20 articles from this search were utilised in this review (See Appendix A). Other references were added to format the introduction, and together with other article leads (including international tick vaccine research and historic publications), this led to a total of 96 references for this review. These also included Meat & Livestock Australia reports as a relevant source of information available on the website (https://www.mla.com.au/research-and-development/search-rd-reports/, accessed on 3 June 2021) and available online vaccine patents.

## 3. Tick Vaccine Research

Tick vaccine research in Australia has been supported since the 1980s for both the cattle tick and the Australian paralysis tick through several groups, see Table 1. 

### 3.1. Cattle Tick

Early studies by Trager published in 1939 demonstrated that exposing guinea pigs to *Dermacentor variabilis* (American dog tick) larvae led to the protection from subsequent infestations, and the serum could be used to protect guinea pigs from infestation [33]. An experiment published in *Nature* in 1979 showed that *Dermacentor andersoni* tick gut extracts were protective in a cattle tick challenge trial [34]. This phenomenon was further investigated using gut extracts from *R. australis* ticks by two research groups in Australia—CSIRO and UQ; see summary in Table 1.

#### 3.1.1. CSIRO Research~1979–2010

Bm86 was isolated from the midgut of *R. australis* ticks by researchers at CSIRO. Initially, experiments demonstrated the in vivo protection from tick challenge using crude extracts, followed by demonstrated protection using native Bm86 protein [21]. Bm86 was patented, and recombinant protein was produced in *E. coli* through a collaboration with a commercial company and was registered following field trials [22,35]. This product was released as TickGARD^®^ in 1994, and a yeast *Pichia pastoris* expressed Bm86 was released in 1995 as TickGARD^®*PLUS*^. The yeast expressed Bm86 was shown to be more immunogenic due to glycan moieties added during protein synthesis in yeast (reviewed by Willadsen in an industry report [23]). This reflected the research undertaken by a Cuban group that used the CSIRO patented information to express Bm86 in *Pichia pastoris*, which was registered as GAVAC^®^ in 1993. GAVAC^®^’s Bm86 antigen is administered as 100 µg per dose and is currently adjuvanted with Montanide ISA 50 V2 oil adjuvant (https://www.cigb.edu.cu/en/product/gavac-2/, accessed on 1 September 2021) TickGARD^®*PLUS*^ was discontinued in Australia by 2010, yet GAVAC^®^ continues to be produced. The main issue for discontinuation of TickGARD^®*PLUS*^ was the need for 3–4 boosts per year, which was not adopted by the extensive Australian beef industry where mustering of cattle is undertaken only once per year. TickGARD^®*PLUS*^ doses were 50 µg [35]; however, the adjuvant used is not publicly available. 

Apart from the boosting requirements for Bm86 vaccines, the other issue associated with Bm86-based vaccines was that it was found not to be cross-protective in several other countries; see Table 2. Table 2 summarises the data compiled by Dr Peter Willadsen in an industry report [23], as well as published studies of efficacies using the GAVAC^®^ Cuban Bm86 product for Columbia and the USA, and the use of local Campo Grande Bm86 sequence tested in trials in Brazil.

During the development of the Bm86 vaccine, other purified mixtures of native tick proteins were also shown to be protective in vaccination trials [47]. Some of these vaccine antigens were further researched by the CSIRO group as novel antigens and/or also as a cocktail vaccine, including Bm86; see Table 3. This notion was suggested following the registration of Bm86 to find extra antigen(s), which may potentially improve the vaccine [34]. Some experimental trials were also undertaken in sheep, which were initially thought to be a cheaper model for the preliminary testing of novel vaccine antigens for cattle. Bm91 was an additional ‘concealed’ antigen isolated in 1994 and was thought to be an angiotensin-converting enzyme-like protein [48]. A combination vaccine including Bm86 and Bm91 was tested in 1996, demonstrating marginal improvement to Bm86 efficacies [49]. Bm91 was later determined to function as a putative carboxypeptidase [50]. A mucin-like glycoprotein was isolated from fractions following the removal of Bm86 and Bm91 and tested as native protein alone and in combination with Bm86 [51]. BMA7 had moderate effects when used alone as a vaccine and was demonstrated to improve the efficacy of Bm86 alone; see Table 3. Vitellin and vitellogenin vaccinations using native proteins produced good efficacies with a 66–68% reduction in tick larvae; however, the corresponding bacterial recombinant GP80 protein did not demonstrate any efficacies [52]. The efficacy of recombinant 5′ nucleotidase in sheep was reported at 73%, which was lower than Bm86 (85%); however, nil efficacy was reported in cattle for 5′ nucleotidase. In addition, vaccination using both 5′ nucleotidase and Bm86 did not demonstrate an improvement to Bm86 vaccination alone tested in cattle trials [53]. These results demonstrated that sheep are not viable model hosts for use in anti-cattle tick vaccine studies and that the application of a dual vaccine does not always lead to increased efficacies [53].

The majority of the above antigens were tested as native proteins and/or as bacterial recombinant proteins. In many cases, the antigen sequences were identified using N-terminal sequencing of the protein fractions under study. Thus, several of the above publications focused on the fractionation and purification of native proteins in preparation for vaccination and also for sequence analysis. It is thus not known if these proteins may be more efficacious if expressed in yeast. 

#### 3.1.2. UQ Research~1988–1999

Research at UQ was undertaken simultaneously as the above research by CSIRO in the 1990s. The researchers demonstrated success with using gut membrane extract and demonstrated protection of *Bos taurus* Hereford cattle from a subsequent challenge; see Table 1. Soluble gut antigens were shown not to be protective [24]. The research did not progress further to produce specific vaccine candidates; however, it published useful immunological studies, such as the demonstrated correlation of IgG1 and complement for protection from tick challenge, which underpinned future research (see Section 3.1.3). The research ceased when the lead scientist at the Department of Parasitology, Dr Joan Opdebeeck, left UQ in the late 1990s.

#### 3.1.3. Cooperative Research Centre for Beef Genetic Technologies (Beef CRC) 2005–2012; Meat & Livestock Australia (MLA) 2014–2021

Due to the above reported variable efficacies of Bm86-based vaccines as well as the need to boost 3–4 times per year, a new research team commenced within the Beef CRC, which included the Queensland Department of Agriculture & Fisheries (QDAF, previously Department of Primary Industries & Fisheries/DPI&F and Department of Employment, Economic Development & Innovation/DEEDI), Murdoch University’s Centre for Comparative Genomics and the US Department of Agriculture as the main research partners. The team employed a reverse vaccinology approach by screening a database of *R. microplus* EST sequences (n = 13,643) and by undertaking gene discovery experiments to identify transcripts from host sensing and feeding of different tick life cycle stages of *R. australis* [54,55,56]. B cell epitopes of selected candidates were screened in ELISAs using serum from tick susceptible and resistant Santa Gertrudis composite breed cattle [57]. Epitopes recognised by resistant cattle sera were tested in an in vitro adult female tick assay by antibody feeding [58]. Mixtures of epitopes showing high efficacies in antibody tick feeding screens were tested in tick challenge cattle trials, and the individual antigens from successful mixtures were subsequently tested in cattle trials, and this testing was completed in 2017 [26]. Nine candidates were subsequently submitted as full patents, and a recent longevity trial demonstrated high increasing efficacies 6 months after the initial dual vaccination using a dual antigen vaccine (MLA report, unpublished). Commercial interest has yet to be gauged by the Intellectual Property owners, and, as such, further specific details cannot be provided in this review.

### 3.2. Paralysis Tick

Early research in the 1920s by Sir Ian Clunies Ross determined the protection from tick paralysis by using crude salivary gland extracts from *I. holocyclus* adult female ticks [59]. Treatment of paralysis using serum from hyperimmune dogs was also demonstrated early last century [60]. The toxin fraction from the salivary glands of adult female ticks was first ascertained by Kaire in 1966 [61]. These seminal studies provided the background for future research, characterising a family of neurotoxins enabled only by developments in modern biotechnologies, such as proteomics and transcriptomics. Although other tick species are known to cause paralysis in other countries, i.e., *Rhipicephalus evertsi evertsi* (lamb paralysis, South Africa), *Argas (Persicargus) walkerae* (poultry paralysis, South Africa), *Dermacentor variabilis* (dog paralysis in the USA), the causative proteins or toxins are yet to be elucidated at the molecular level [62,63,64].

#### 3.2.1. CSIRO Research 1980s

Several studies were undertaken by Dr Bernard Stone and colleagues at CSIRO, including attempts to isolate the causative toxin(s) [17,65,66]. These studies concluded that the toxin was in the size range of 40–80 kDa as representative of the toxic fractions dissected from whole ticks or adult female salivary glands. This result reflected research identifying toxin fractions from *R. evertsi evertsi* and *A. (P.) walkerae*, which were reported to be similar in size [64]. As mentioned in Table 1, this group developed assays for screening toxin fractions, which underpinned future research in this field. Although partially purified toxin extracts from *I. holocyclus* treated with glutaraldehyde showed good anti-toxin responses [67], the practicality of preparing native tick material was not feasible for vaccine development. The term holocyclotoxin was coined by Stone and co-workers, and although mentioned in earlier publications, the 1986 publication used the term officially in the journal’s title [67]. 

#### 3.2.2. University of Technology Sydney (UTS) Research 1990s

The above research of Stone and colleagues overlapped with research undertaken by Associate Professor Kevin Broady and colleagues at UTS, who were also attempting to isolate the causative paralysing holocyclotoxins. Three holocyclotoxins HT-1, HT-2 and HT-3 were isolated from rat brain synaptosomes (pinched off nerve terminals), which bind the neurotoxins in a temperature-dependent manner [31]. These polypeptides had a molecular mass of 5 kDa, and the HT-1 gene was isolated using PCR technologies in 1992. A recombinant HT-1 was produced and shown to be immunogenic; however, the native sequences of HT-2 and HT-3 were not characterised. It was not certain whether the genes were distinct or whether they were post-translational modified forms of HT-1 (reviewed by Masina and Broady 1999) [32]. Thus, as mentioned above, the large HT family had yet to be characterised at the molecular level.

#### 3.2.3. UQ Research 2000s

Approximately 17 years following the initial description of HT-1 and research program commenced at UQ, which aimed to sequence the transcriptome of *I. holocyclus*, gut and salivary gland organs from paralysed companion animals were collected from Veterinary Surgeries from the broader Brisbane area. Two different pooled organ samples were sequenced from guts and salivary glands using Illumina short-read technology, and a family of ~19 HTs (including the initial HT-1 described above) were characterised with amino acid identities ranging between 38% and 99% [16]. A proof-of-concept dog trial of a cocktail of HTs was undertaken and showed protection from *I. holocyclus* tick challenge (unpublished). Commercialisation through a local tick antiserum company in Australia is currently underway (author corporate knowledge). 

### 3.3. Australian Researchers (Deceased) Who Made Significant Impacts into Tick Vaccine Research

During the process of this review, it was clear that several researchers in Australia were responsible for seminal studies, which led to the development of the current vaccines against cattle ticks and paralysis ticks. These researchers were mostly employed by CSIRO, and their research legacy paved the way for tick vaccine development in Australia. Each of these three researchers was dedicated to science until they passed with phenomenal legacies.

#### 3.3.1. Sir Dr Ian Clunies Ross (1899–1959)

Dr Ian Clunies Ross completed a bachelor’s degree in veterinary science (B.V.Sc.) at the University of Sydney in 1921. Clunies Ross spent 1921 as a temporary lecturer in veterinary anatomy at the university, and in 1922, he was appointed as a Walter and Eliza Hall research fellow. In England, he studied parasites at the Molteno Institute, Cambridge and at the London School of Tropical Medicine. In 1925, he resumed research on parasites and undertook some part-time teaching at the University of Sydney’s Veterinary School (https://adb.anu.edu.au/biography/clunies-ross-sir-william-ian-9770, accessed on 1 September 2021) He was employed as a parasitologist at the Council for Scientific and Industrial Research (CSIR—later CSIRO) in the mid1920s. His pioneering research in the paralysis tick laid the foundation for ongoing research into the toxic fractions of the Australian paralysis tick *I. holocyclus*, as described above [59,68,69]. Sir Dr Ian Clunies Ross’s publications have been previously summarized [70]. The review by Gordon was included in a special issue dedicated to Sir Dr Ian Clunies Ross in Volume 44, Issue 10 (October 1968) of the *Australian Veterinary Journal*. His research interests included the hydatid parasite *Echinococcus granulosis* and the liver fluke *Fasciola hepatica*, with a strong research focus associated with the development of livestock antihelmintic treatments (see citations in Gordon 1968 [70]). During his research career, he published approximately 70 research publications, including surveys and records, immunity and resistance, pathogenesis, antihelmintics and the application of control measures across many livestock parasite species [70]. In 1954, Clunies Ross was knighted and appointed CMG (Order of St Michael and St George) and became a Foundation Fellow of the Australian Academy of Science. He also held administrative leadership roles as the Chairman of CSIRO from 1946 (formerly CSIR) until his passing in 1959. 

#### 3.3.2. Dr Bernard F. Stone (1926–2005) (Marian Schneid, Personal Communication)

In 1954, Bernard Stone joined CSIRO, Division of Entomology, working on cattle tick biocide resistance, and authored his first article about parthenogenesis in the cattle tick in 1963 [71]. He completed his BSc and Master of Science (MPhil) at UQ by 1966 and his PhD with the University of Western Ontario (Canada) by 1968. Dr Stone was awarded a UQ Doctor of Science in 1982. As a researcher, he was quite prolific with 90 research publications (Marian Schneid, personal communication) with a strong impact on the development of *I. holocyclus* toxoid and the standardisation of tick anti-toxin production in Australia for treating paralysis in domestic pets and livestock [17,27,29,67,72]. His publications also focused on *R. australis* drug resistance [73,74,75]. In 1994, he was awarded an Order of Australia for his contributions to science, environment, youth and scouting. Following retirement, he led a consultancy business evaluating commercial antitoxins for tick paralysis until his passing in 2005. 

#### 3.3.3. Dr David H. Kemp (1940–2007) (Dr Anne Kemp, Personal Communication)

After completing his PhD on *Ixodes ricinus* ticks on sheep in Scotland at The University of Edinburgh, Dr Kemp was employed by CSIRO in Australia with the Division of Animal Health. Dr Kemp had an extensive career in cattle tick research with CSIRO, spanning almost 40 years, and he was one of the main protagonists developing the CSIRO TickGARD vaccine [18,21,76]. He also collaborated with Allen, who initially discovered the protective properties of *Dermacentor variabilis* tick gut extracts in 1979 [34,77]. Specifically, together with his technician Joanne Gough, he noted that tick-resistant breeds of cattle carried ticks, which appeared as hemorrhages internally (red legs) (Dr Anne Kemp, personal communication). He was a tick biologist with many interests: improving cattle resistance to ticks, tick histological structure, the taxonomy of Australian ticks (his collection is currently within CSIRO’s Australian Insect Collection, Canberra; Australian National Insect Collection-CSIRO) and leading ACTEST—tick acaricide testing facility (Note by author: ACTEST was transferred to QDAF in the 2000s). Approximately 60 of Dr Kemp’s publications are recorded in NCBI’s PubMed (searched 12/07/2021), mostly associated with *R. australis* (as *B. microplus*), with additional research contributions to *I. holocyclus* associated with livestock infestations [78,79,80], and other tick/ectoparasite species [81,82]. Research collaboration between Dr Kemp and Dr Stone (as members of different Divisions in CSIRO) was demonstrated with the development of an in vitro feeding system for *I. holocyclus* toxin collection [28]. 

Dr Kemp also contributed to the commencement of the new Beef CRC/MLA research program (Section 3.1.3) (during his retirement) through his initial suggestions to examine ‘frustrated ticks’ (ticks in bags sensing the cattle host) in vaccine candidate discovery research. This led to vaccine candidates, which were included in the reverse vaccinology approach by identifying transcripts produced when sensing cattle [55,56]; see Table 1.

### 3.4. Current Status of Global Cattle Tick Vaccine Research

Several cattle tick vaccine candidates have been patented, and these candidates were recently reviewed by Tabor [83] and will not be re-reviewed here. Currently, CATVAC (cattle tick vaccine consortium) is testing several of these antigens in trials in Morocco funded through the Bill & Melinda Gates Foundation in an effort to identify a globally applicable cattle tick vaccine [43]. These include:

#### 3.4.1. Bm86 and Subolesin

Bm86 and Subolesin as a dual vaccine (100 µg per antigen), was delivered in a double emulsion to cattle using three boosts, which has shown a 97% reduction in tick numbers [84] following challenge with a Mexican *R. microplus* and a Texan *R. annulatus* strains. The Bm86 sequence was based on the Mexican *R. microplus* sequence; thus, the experiment represented a homologous Bm86 challenge. Subolesin (ortholog of ankyrin in ticks) was identified as a tick protective antigen in *I. scapularis* ticks [85], and the *R. microplus* homologue was shown to protect ~60% efficacy in cattle [86]. It is currently not known if the variation in Bm86 efficacies (Table 2 above) will impact the commercialisation of this vaccine.

#### 3.4.2. Aquaporin

Aquaporin, a water-conducting channel, was suggested as a vaccine candidate for the blood fluke *Schistosoma japonicum* with several predicted immunogenic epitopes and an integral transmembrane structure [87]. RNA interference studies demonstrated a strong knockdown effect on the *R. microplus* aquaporin homologue, and a 73% efficacy was confirmed following cattle trial vaccination using three boosts of 100 µg recombinant protein in a Montanide ISA 61 VG adjuvant and tick challenge [88].

#### 3.4.3. 60S Acidic Ribosomal Peptide P0

P0 has demonstrated 96% efficacy against *R. microplus* in cattle trials in Cuba using four doses of 250 µg of peptide in VG Montanide 888 adjuvant prior to challenge [89]. P0 was previously shown to be lethal in RNA interference studies in a tick *Haemaphysalis longicornis* [90]. It is likely that the peptide production of a bovine vaccine has stalled the commercialisation of this candidate.

#### 3.4.4. Ferritin-2

Ferritin-2 has shown promise as a vaccine candidate (three doses of 100 µg prior to tick challenge) in *I. ricinus* (rabbits with Incomplete Freund’s adjuvant) and also *R. microplus* (cattle with Montanide ISA 50V adjuvant) with 96% and 64% efficacies, respectively [91]. The candidate was discovered when studying iron homeostasis in *I. ricinus* was found to be essential in tick feeding. 

## 4. Conclusions

It is clear that, in particular, due to the commercialisation of the cattle tick Bm86 protein, vaccination against ticks is a feasible control option [34]. The research by CSIRO described the impact of Australian research on the development of anti-tick vaccines. For *I. holocyclus*, a vaccine alternative to companion animal chemical treatments is also a future option for this unique Australian ectoparasite. This author notes that due to the focused impact of this tick species on the east coast of Australia, international animal health companies deem that commercialisation is not economically feasible. With the shortcomings associated with *R. microplus*/*R. australis* delivery, both with poor global efficacies and the need for multiple boosting, much research has focused on identifying alternative vaccine candidates, as described above. What is disconcerting is the slow commercialisation of promising new cattle tick vaccine candidates, a topic recently reviewed by de la Fuente and Estrada-Pena in 2019 [92]. Indeed, of the anti-cattle tick vaccines described in this review, only the Bm86 antigen vaccine GAVAC^®^ is currently commercially available. De la Fuente and Estrada-Pena discuss the challenges associated with the fact that the application of generic chemicals is a more attractive alternative to specific vaccines (confirmed by the author’s personal experience with *R. australis* vaccine commercialisation as companies strive to develop generic chemicals as alternatives to vaccines); the expense of novel adjuvant delivery systems for livestock; the challenge to produce anti-tick vaccines that can be applied across multiple tick species; and finally, the ability to control tick-vectored pathogens simultaneously [92]. It remains to be seen if new products will become available to the market. Vaccines will always provide a safe alternative to chemicals by reducing environmental residues and costs in the administration of these chemicals. 

## Figures and Tables

**Table 1 vaccines-09-01030-t001:** Summary of tick vaccine research in Australia—the Australian cattle tick *Rhipicephalus australis* and the Australian paralysis tick *Ixodes holocyclus*.

Target Species	Group	Outputs	References
*Rhipicephalus australis*	CSIRO	Demonstrated protection from challenge using crude gut extracts; Isolation of gut protein Bm86 and demonstrated protection from challenge using native protein; Patent filing in 1985/1986, followed by commercial development and release of TickGARD^®^ (1994) and TickGARD^®*PLUS*^ (1995)	[21,22,23]
	UQ	Demonstrated protection from challenge using midgut membrane extracts; Demonstration that gut membrane extracts are protective compared to soluble antigens; Showed that IgG1 and complement antibodies correlated with protection	[24,25]
	Beef CRC/MLA	Reverse vaccinology pipeline was developed to identify novel tick vaccine antigens; Full patents filed February 2018; Antigens under Intellectual Property protection, data unpublished	[7,26]
*Ixodes holocyclus*	CSIRO	Developed paralysis tick toxoid for anti-toxin activity demonstrated in rabbits; Artificial feeding methods for the collection of paralysing toxin; Development of a suckling mouse toxin/anti-toxin assay; Protection shown in dogs	[15,27,28,29,30]
	University of Technology Sydney	Isolated three polypeptides bound to synaptosomes; Successfully sequenced and expressed holocyclotoxin-1 (HT-1) in *E. coli*	[31,32]
	UQ	Transcriptome sequence database prepared from feeding adult female tick guts and salivary glands; Based on HT-1 homology, a diverse family of 19 HTs was identified; Proof of concept dog trial with a cocktail of synthetic HTs; Full patent submitted October 2017; Data unpublished	[16]

**Table 2 vaccines-09-01030-t002:** Summary of Bm86 efficacies relative to TickGARD^®^ strain *R. australis* Yeerongpilly—modified from Willadsen 2008 [23], including additions for the USA, Columbia and Brazil.

*R. microplus* Isolate	Sequence Difference	Efficacy	References
*R. australis* Indooroopilly Y (Australia)	0%	89%	[36]
*R. microplus* Tuxpan (Mexico)	5.7%	51%	[37]
*R. microplus* Mora (Mexico)	8.6%	58%	[37]
*R. microplus* (Mexico)	3.3%	89%	[38] ^1^
*R. microplus* (Argentina)	1.6%	10%	[37]
*R. microplus* Camcord GAVAC strain (Cuba)	0.2%	84%	[39,40,41]
*R. microplus* (Columbia)	unpublished	81%	[42]
*R. microplus* Texas (USA)	10.2%	50–65%	[43,44]
*R. microplus* Campo Grande (Brazil)	3.5%	31%	[45,46]

^1^ Cited by Willadsen [23]; article not accessible for this review.

**Table 3 vaccines-09-01030-t003:** Anti-tick antigens developed and tested by CSIRO after Bm86 registration.

Antigen Description	Testing Regime	Published Efficacy	References
Bm91/Angiotensin-converting enzyme-like/Carboxypeptidase	Bm86 + Bm91, tested in cattle	7% reduction in egg viability	[48,49,50]
BMA7/Mucin-like glycoprotein	BMA7; Bm86+BMA7, tested in cattle	21% reduction in egg weights; doubled efficacy when mixed with Bm86	[51]
GP80/VIT87 Vitellin/Vitellogenin	Tested in sheep	66–68% reduction in tick larvae (native protein); 0% efficacy (recombinant)	[52]
5′ nucleotidase/Apyrase	5′ nucleotidase; Bm86 + 5′ nucleotidase, tested in sheep and cattle	No efficacy (cattle)	[53]

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
