# Peer review of "A Review of Australian Tick Vaccine Research"

_vaccines, 2021, doi:10.3390/vaccines9091030_

Round 1
Reviewer 1 Report
It is a very informative and well-written review paper on the subject. Few minor errors are there as follows.
- Italicize the scientific name of ticks in the abstract, and line 301~ 319 etc.
- Line 61: 19th -> 20th
Author Response
- Italicize the scientific name of ticks in the abstract, and line 301~ 319 etc.
These have been corrected.
- Line 61: 19th -> 20th
This has been corrected.
Reviewer 2 Report
This paper provided an overview of the work/research from Australian groups regarding tick vaccines. The paper is well written and very interesting, however, the scope will be limited to mainly that group. This review might be more suitable for a local journal? It would have been more valuable if the review covered the development of tick vaccines with all the major contributions, not only those from Australia. It is only due to the local interest that I rejected it.
with some minor suggestions:
Line 9: CSIRO-spell out in full, this is an acronym.
Line 311: R. microplus-Italics for species names, check the whole document for consistency please.
Line 315: Haemaphysalis longicornis-Italics
Line 319: I. ricinus-Italics
Author Response
This paper provided an overview of the work/research from Australian groups regarding tick vaccines. The paper is well written and very interesting, however, the scope will be limited to mainly that group. This review might be more suitable for a local journal? It would have been more valuable if the review covered the development of tick vaccines with all the major contributions, not only those from Australia. It is only due to the local interest that I rejected it.
Response: This reviewer perhaps is not aware that this is a ‘Vaccines Development in Australia’ special issue thus this comment has been not been addressed.
with some minor suggestions:
Line 9: CSIRO-spell out in full, this is an acronym.
This has now been spelt out in the Abstract.
Line 311: R. microplus-Italics for species names, check the whole document for consistency please.
The whole document has been rectified.
Line 315: Haemaphysalis longicornis-Italics
Rectified
Line 319: I. ricinus-Italics
Rectified.
Reviewer 3 Report
The MS is a retrospective analysis of Australian tick-related vaccine development. It is a well-structured review, with important details highlighted, still easy to read, while scientifically sound. I strongly suggest its acceptance.
A small issue below:
L21 Ticks were evolved far more earlier than humans, while humans are also animals… This is a highly sensational statement, not to be used in such a review, please revise.
Author Response
L21 Ticks were evolved far more earlier than humans, while humans are also animals… This is a highly sensational statement, not to be used in such a review, please revise.
Sentence has been revised to: ‘Ticks are hematophagous (blood sucking) arthropods which have evolved with animal hosts.’
Reviewer 4 Report
The manuscript presents a review of tick vaccine research in Australia. The subject is interesting and reports new data obtained from family members of some of the top researchers of this subject (I would suggest a review to the way to reference these direct contributions, since they are presented in different forms). However, the text is somewhat confusing and sometimes difficult to follow.
One example regards the Rhipicephalus (Boophilus) microplus/australis tick; tick taxonomy is not the focus of this review, so the state-of-the art regarding this tick taxonomy should be described in the Introduction, and the author should choose the actual tick designation, that should be used from this point on. To go back and forward using different ways to designate this tick along the text is confusing.
Another aspect is that in a review about anti-tick vaccines, no information regarding the antigens research is presented. Important characteristics to choose antigens, their potential accessibility to antibodies and these to tick by ingestion during feeding and subsequent impairment of a physiological function of importance to the tick should be presented in the Introduction.
A throughout text review to enable a fluent reading could provide significant improvements to the manuscript.
Specifically, I have the following comments:
- Line 9: CSIRO – abbreviation should be identified at the first time it is referred. Other used abbreviations also need to follow this procedure as University of Queensland (UQ), in line 152, and removed from line 222.
- Line 10: “ …to the development and or precursors of vaccines…”, this sentence is confusing or misspelt. Do you mean: “… to the development of vaccines and/or vaccine precursors…”?
- Line 11: the species name “Rhipicephalus microplus/australis” – the species name should be in italic and in the abstract the species designation should be the actually recognized so, Rhipicephalus australis.
The issues regarding the species renaming and previous designation are not the main manuscript focus and should (in my opinion) only be presented in the manuscript text (not in the abstract).
- Line 14: Ixodes holocyclus – should be in italic;
- Lines 30, 46, 99… : Numbers 1 to 9 are written in full (MDPI - Vaccines Guidelines for Authors)
- Lines 40 and 42: the adjective/adverb worldwide is spelled as one word
- Line 61: 19th Century? – 20th Century, right? Also, following MDPI - Vaccines Guidelines for Authors, “The 'th' in 19th or 20th should NOT be written in superscript.”
- Line 63: PubMed database search data should be presented
- Line 116: “…the other issue of associated with…”, remove of
- Line 141-142: The sentence should be changed to a clearer statement about the inefficacy of the use of 5’-nucleotidase as antigen in an anti-tick vaccine, since the results obtained in sheep were not confirmed in cattle, and even in sheep, protection was less then found with Bm86. All data from Hope et al (2010) questions the use of sheep as model host in anti-tick vaccine studies, advising that sheep should not be use in these studies. The text (lines 141-145) could be clearer.
- Line 206: “Stone and workers” suggestion to replace as “Stone and colleagues or Stone and co-workers”
- Line 208-209: I do not understand the need to present this sentence, or I am not following what you mean… If the designation of holocyclotoxin was only officially recognized after 1986, so in some of the previous research studies these molecules are referred simple as toxins. If this is it, I think this last sentence becomes redundant and confusing.
- Lines 301-319: All species names should be corrected to italic typeface
- Line 321: “It is clear that that in particular”, delete extra that…
- Table 1: table formatting could be clearer and easier to read. If the first chosen column is the Research group, the outputs should be presented for the group in both tick target species. If the author believes that the outputs should be presented regarding the tick species this column should the first of the table
- Table 3: 5’ nucleotidase please add the antigen description
Reference section should be reviewed to address MDPI - Vaccines Guidelines
Author Response
The manuscript presents a review of tick vaccine research in Australia. The subject is interesting and reports new data obtained from family members of some of the top researchers of this subject (I would suggest a review to the way to reference these direct contributions, since they are presented in different forms). However, the text is somewhat confusing and sometimes difficult to follow.
Response: I have reviewed these sections and Clunies Ross’s early publications were published in 1968, however for Dr Stone and Dr Kemp I had to contact relatives for exact dates and further information. For Dr Stone – the family had a CV, for Dr Kemp, the family did not have a CV thus I undertook a Pubmed search to confirm the number of his contributions. For both Dr Stone and Dr Kemp, the families provided biographies to ensure the information used here was correct, however it was difficult to find reliable biographical information for Clunies Ross. The family members and listed in the acknowledgments. I have added names to the titles of the sections and I reviewed the text to improve clarity and consistency of information for all three researchers. I found a biographical website for Clunies Ross and added the extra biographical information about his qualifications and experience prior to 1926.
One example regards the Rhipicephalus (Boophilus) microplus/australis tick; tick taxonomy is not the focus of this review, so the state-of-the art regarding this tick taxonomy should be described in the Introduction, and the author should choose the actual tick designation, that should be used from this point on. To go back and forward using different ways to designate this tick along the text is confusing.
Response: the ‘controversial name changes text’ has been moved to the introduction. There it describes how for the review the name R. australis is used when referring to Australian R. microplus/B. microplus research.
Another aspect is that in a review about anti-tick vaccines, no information regarding the antigens research is presented. Important characteristics to choose antigens, their potential accessibility to antibodies and these to tick by ingestion during feeding and subsequent impairment of a physiological function of importance to the tick should be presented in the Introduction.
Response: This is not really the scope of this particular review- in many instances the mechanisms are not well known apart from perhaps Bm86. However, a new paragraph has been added to the Introduction:
‘Approaches for the development of anti-tick vaccines have been quite variable particularly for R. microplus and R. australis globally. One approach has been to target gut associated ‘concealed antigens’ which are usually not recognized by the host during natural infestation, however when used as a vaccine antigen the host’s antibodies attack the tick gut disrupting the life cycle [18]. One disadvantage is that these vaccines require several boosts each year as tick challenge does not boost host responses to these types of antigens. Other vaccine approaches have focused on secreted salivary antigens which conversely are assumed to be recognized by the host during natural challenge and thus may potentially boost vaccine responses [19]. This is particularly relevant for tick species whose tick stages feed on their hosts for shorter periods i.e. I. holocyclus [17, 20], however these approaches have also been applied in the discovery of R. microplus antigens [19].’
A throughout text review to enable a fluent reading could provide significant improvements to the manuscript.
Response: Other edits have been made to manuscript which hopefully address this reviewer’s concerns.
Specifically, I have the following comments:
Line 9: CSIRO – abbreviation should be identified at the first time it is referred. Other used abbreviations also need to follow this procedure as University of Queensland (UQ), in line 152, and removed from line 222.
Response: this has been rectified (CSIRO) – see reviewer 2. UQ has also been rectified throughout the document for consistency.
Line 10: “ …to the development and or precursors of vaccines…”, this sentence is confusing or misspelt. Do you mean: “… to the development of vaccines and/or vaccine precursors…”?
Response: This statement is referring to the research where ‘tick extracts’ were used to demonstrate vaccine protection thus the statement has been modified to read: ‘… to the development of vaccines and/or vaccine precursors (such as crude extracts)’
Line 11: the species name “Rhipicephalus microplus/australis” – the species name should be in italic and in the abstract the species designation should be the actually recognized so, Rhipicephalus australis.
Response: ‘microplus/’ has been deleted.
The issues regarding the species renaming and previous designation are not the main manuscript focus and should (in my opinion) only be presented in the manuscript text (not in the abstract).
Line 14: Ixodes holocyclus – should be in italic;
Rectified.
Lines 30, 46, 99… : Numbers 1 to 9 are written in full (MDPI - Vaccines Guidelines for Authors)
These have been rectified.
Lines 40 and 42: the adjective/adverb worldwide is spelled as one word
Rectified
Line 61: 19th Century? – 20th Century, right? Also, following MDPI - Vaccines Guidelines for Authors, “The 'th' in 19th or 20th should NOT be written in superscript.”
Response: this has been rectified.
Line 63: PubMed database search data should be presented
Response: A supplementary table has been added to the manuscript with this information (Table S1) and edits to the review methodology updated with further information to improve clarity.
Line 116: “…the other issue of associated with…”, remove of
Rectified
Line 141-142: The sentence should be changed to a clearer statement about the inefficacy of the use of 5’-nucleotidase as antigen in an anti-tick vaccine, since the results obtained in sheep were not confirmed in cattle, and even in sheep, protection was less then found with Bm86. All data from Hope et al (2010) questions the use of sheep as model host in anti-tick vaccine studies, advising that sheep should not be use in these studies. The text (lines 141-145) could be clearer.
Response: This section has been re-worded as follows:
The efficacy of recombinant 5’ nucleotidase in sheep was reported at 73% which was lower than Bm86 (85%), however nil efficacy was reported in cattle for 5’ nucleotidase. In addition, vaccination using both 5’ nucleotidase and Bm86 did not demonstrate an improvement to Bm86 vaccination alone in cattle trials [53]. These results demonstrated that sheep are not viable model hosts for use in anti-cattle tick vaccine studies, and that the application of a dual vaccine does not always lead to increased efficacies [53].
Line 206: “Stone and workers” suggestion to replace as “Stone and colleagues or Stone and co-workers”
Response: this has been edited to ‘Stone and co-workers’
Line 208-209: I do not understand the need to present this sentence, or I am not following what you mean… If the designation of holocyclotoxin was only officially recognized after 1986, so in some of the previous research studies these molecules are referred simple as toxins. If this is it, I think this last sentence becomes redundant and confusing.
Response: the statement has been deleted.
Lines 301-319: All species names should be corrected to italic typeface
These have been rectified.
Line 321: “It is clear that that in particular”, delete extra that…
Deleted
Table 1: table formatting could be clearer and easier to read. If the first chosen column is the Research group, the outputs should be presented for the group in both tick target species. If the author believes that the outputs should be presented regarding the tick species this column should the first of the table
Response: Table 1 has been rectified to have the tick species in the first column.
Table 3: 5’ nucleotidase please add the antigen description
Response: The term ‘apyrase’ has been added to the antigen description in Table 3.
Reference section should be reviewed to address MDPI - Vaccines Guidelines
Response: the references have been reviewed and edited.
Reviewer 5 Report
In this review article, Dr. Ala Tabor summarized the published research related to tick vaccine efforts which were primarily undertaken in Australia. While the article is valuable in recuperating old and current literature related to the tick vaccine research, it needs several improvements. The following are a few specific comments.
1) It will be valuable to add more detailed data (as a Table format) to summarize the vaccine formulations and doses, and include adjuvants used in defining the protection.
2) Also valuable is to provide details regarding hosts where a vaccine is applied and with a timeline of such vaccine application. Further, it is important to provide details about a vaccine’s current application status (active or not in use). These details will make the review much more meaningful.
3) Several statements are not appropriate and/or are nonscientific. Here are few examples and recommendations.
4) I recommend deleting the following first sentence from the abstract; “Tick vaccine research in Australia has demonstrated leadership worldwide.”
5) The following first sentence in the introduction section is not scientific: “Ticks are ‘mini vampires’ which have evolved with their human and animal hosts.” I recommend deleting it as vampires are mythical creatures.
6) The word ‘respectively’ is not needed on line 28 of introduction section.
7) It is not clear which animals the author is referring to in the sentence on lines 60-61.
8) Delete the word ‘strongly’ from the sentence in lines 71-72.
9) Delete the words ‘Note that’ from line 75.
10) I recommend editing the sentence on lines 108-109 to the following: The yeast expressed Bm86 was shown to be more immunogenic due to glycan moieties added during the protein synthesis in yeast.
11) Most of the text was presented in past tense. The preferred format is that all prior published reports should be presented in present tense.
12) Delete one ‘that’ (where two that’s are present) from the sentence on line 321.
Author Response
In this review article, Dr. Ala Tabor summarized the published research related to tick vaccine efforts which were primarily undertaken in Australia. While the article is valuable in recuperating old and current literature related to the tick vaccine research, it needs several improvements. The following are a few specific comments.
1) It will be valuable to add more detailed data (as a Table format) to summarize the vaccine formulations and doses, and include adjuvants used in defining the protection.
Response – It is unclear which vaccine formulations or for which vaccines the reviewer is referring to exactly. I am not sure if pertinent to the review as for experimental vaccines – non commercially available adjuvants are usually used. Also, for several commercial vaccines – the adjuvant is proprietary and as such is not published or available to add to this review e.g. TickGARD. In addition to ‘doses’ this is difficult when crude extracts (salivary gland material as an example for Ixodes holocyclus). Nonetheless these statements have been added to page 4: ‘GAVAC®’s Bm86 antigen is administered as 100 µg per dose and is currently adjuvated with Montanide ISA 50 V2 oil adjuvant (https://www.cigb.edu.cu/en/product/gavac-2/).’ And ‘TickGARD®PLUS doses were 50µg [34] however the adjuvant used is not publicly available.’
See also point 2 below for updated text associated with doses/formulations/adjuvants for non-Bm86 cattle tick antigens where this information is available.
2) Also valuable is to provide details regarding hosts where a vaccine is applied and with a timeline of such vaccine application. Further, it is important to provide details about a vaccine’s current application status (active or not in use). These details will make the review much more meaningful.
Response to ‘Also valuable is to provide details regarding hosts where a vaccine is applied and with a timeline of such vaccine application.’ –Hosts have been added to vaccine studies where it is not clear throughout the manuscript. I am uncertain re: ‘timeline of such vaccine application’ is pertaining to, however I have also added text to describe dose/boost numbers and adjuvants used where feasible throughout the manuscript where it was previously lacking.
Response (vaccine’s current status): The only vaccine in this review currently in use commercially is GAVAC® – this statement has been added to the Conclusions section: ‘Indeed, of the anti- cattle tick vaccines described in this review, only the Bm86 antigen vaccine GAVAC® is currently commercially available.’
3) Several statements are not appropriate and/or are nonscientific. Here are few examples and recommendations:
4) I recommend deleting the following first sentence from the abstract; “Tick vaccine research in Australia has demonstrated leadership worldwide.”
Response: This review is about vaccines in Australia thus I do not wish to delete – I have modified thus to: ‘Tick vaccine research in Australia has demonstrated leadership worldwide through the development of the first anti-tick vaccine in the 1990s.’
5) The following first sentence in the introduction section is not scientific: “Ticks are ‘mini vampires’ which have evolved with their human and animal hosts.” I recommend deleting it as vampires are mythical creatures.
Response: This has been modified to: ‘Ticks are hematophagous (blood sucking) arthropods which have evolved with animal hosts.’ See also response to reviewer 3 above.
6) The word ‘respectively’ is not needed on line 28 of introduction section.
Deleted
7) It is not clear which animals the author is referring to in the sentence on lines 60-61.
Response: the word ‘canine’ has been added to this sentence.
8) Delete the word ‘strongly’ from the sentence in lines 71-72.
Deleted
9) Delete the words ‘Note that’ from line 75.
Delete
10) I recommend editing the sentence on lines 108-109 to the following: The yeast expressed Bm86 was shown to be more immunogenic due to glycan moieties added during the protein synthesis in yeast.
Edited as suggested
11) Most of the text was presented in past tense. The preferred format is that all prior published reports should be presented in present tense.
Response: The author has searched the MDPI guidelines for grammar and tenses and there is no ‘preference’ for present tense as the reviewer has suggested. This is a review and as such – most is ‘past tense’ as the manuscript does not describe recent experiments as would be the case if this was a research paper rather than a review. See: MDPI | English editing guidelines for authors
Please also see these generic guidelines: Verb tenses in scientific manuscripts (infographic) - International Science Editing
No changes were thus made in reference to this comment.
12) Delete one ‘that’ (where two that’s are present) from the sentence.
Deleted.
Round 2
Reviewer 2 Report
Paper is interesting to read and provide a nice overview for work done in the specific area.
Author Response
Thank you
Reviewer 4 Report
I believe that with the changes made by the author the review manuscript version was improved and is now acceptable for publication considering only a few specific comments (below).
Regarding my suggestion to add information regarding the antigens research, I am aware that this is not really the scope of this particular review, however in a way to atract a broader sprectrum of readers, that might not be so informed about vaccines development in ticks, this information could be important.
Specifically:
Line 72: please use "5%"
Line 94: delete first extra (
Line 333: number 3, should be written in full
Line 337: replace Ixodes scapularis by I. scapularis
Author Response
Thank you
These edits have all been done.
Reviewer 5 Report
No more suggestions
Author Response
Thank you